# Enhancing Learning with Label Local Differential Privacy by Vector Approximation

**Puning Zhao, Jiafei Wu, Zhe Liu**[*]
Zhejiang Lab
Hangzhou, Zhejiang, China
`{pnzhao, wujiafei, zhe.liu}@zhejianglab.com`

**Li Shen**
Shenzhen Campus of Sun Yat-sen University
Shenzhen, Guangdong, China
`mathshenli@gmail.com`

**Zhikun Zhang**
Zhejiang University
Hangzhou, Zhejiang, China
`zhikun@zju.edu.cn`

**Rongfei Fan**
Beijing Institute of Technology
`fanrongfei@bit.edu.cn`

**Le Sun**
Nanjing University of Information Science and Technology
`lesun1@nuist.edu.cn`

**Qingming Li**
Zhejiang University
`liqm@zju.edu.cn`

## Abstract

Label differential privacy (DP) is a framework that protects the privacy of labels in training datasets, while the feature vectors are public. Existing approaches protect the privacy of labels by flipping them randomly, and then train a model to make the output approximate the privatized label. However, as the number of classes $K$ increases, stronger randomization is needed, thus the performances of these methods become significantly worse. In this paper, we propose a vector approximation approach for learning with label local differential privacy, which is easy to implement and introduces little additional computational overhead. Instead of flipping each label into a single scalar, our method converts each label into a random vector with $K$ components, whose expectations reflect class conditional probabilities. Intuitively, vector approximation retains more information than scalar labels. A brief theoretical analysis shows that the performance of our method only decays slightly with $K$. Finally, we conduct experiments on both synthesized and real datasets, which validate our theoretical analysis as well as the practical performance of our method.

## 1 Introduction

Differential privacy (DP) (Dwork et al., 2006) is an effective approach for privacy protection, and has been applied extensively (Erlingsson et al., 2014; Ding et al., 2017; Tang et al., 2017; Near, 2018; Zhang et al., 2021). However, in supervised learning problems, the original definition of DP can be stronger than necessary, resulting in unnecessary sacrifice of utility. In particular, sometimes it is reasonable to assume that features are publicly available, while only labels are highly sensitive and need to be protected, such as recommender systems (McSherry & Mironov, 2009), computational advertising (McMahan et al., 2013) and medical diagnosis (Bussone et al., 2020). These scenarios usually use some basic demographic information as features, which are far less sensitive than the labels. Under such background, label DP has emerged in recent years (Ghazi et al., 2021;

---
[*]Corresponding author

Malek Esmaeili et al., 2021; Esfandiari et al., 2022; Tang et al., 2022; Cunningham et al., 2022), under which the output is only required to be insensitive to the replacement of training labels.

There are several existing methods for learning with label DP, such as randomized response (Warner, 1965), RRWithPrior (Ghazi et al., 2021) and ALIBI (Malek Esmaeili et al., 2021). These methods flip training labels $Y$ randomly to privatized labels $\tilde{Y}$ to satisfy the privacy requirement. However, a common issue is that the performances decrease sharply with the number of classes $K$. Intuitively, with $K$ increases, it is inevitable for the privatized label to be randomly selected from a large number of candidates, thus the probability of maintaining the original label $P(\tilde{Y} = Y)$ is small. It is possible to design some tricks to narrow down the candidate list, such as RRWithPrior (Ghazi et al., 2021), which picks the candidates based on their prior probabilities. However, the performances under $\epsilon$-label DP with small $\epsilon$ are still far from the non-private baseline. From an information-theoretic perspective, a single scalar can only convey limited information (Cuff & Yu, 2016; Wang et al., 2016; Acharya et al., 2024). Therefore, as long as the label is transformed into a scalar, with increasing $K$, it is increasingly unlikely to maintain the statistical dependence between the original label and the privatized one, thus the model performance drops with $K$.

In this paper, we propose a *vector approximation* approach to solve the multi-class classification problem with large number of classes $K$ under label DP. To satisfy the privacy requirement, each label is transformed into a random vector $\mathbf{Z} = (Z(1), \ldots, Z(K)) \in \{0, 1\}^K$, and the transformation satisfies $\epsilon$-label DP. Intuitively, compared with scalar transformation, conversion of labels to multi-dimensional vectors preserves more information, which is especially important with large $K$, thus our method achieves a better performance.

We then move on to a deeper understanding of our new proposed approach. For a given feature vector $\mathbf{x}$, let $\Delta(\mathbf{x})$ be the maximum estimation error of regression function $\eta_k(\mathbf{x})$ over $k = 1, \ldots, K$. Our analysis shows that as long as $\Delta(\mathbf{x})$ is smaller than the gap between the largest and second largest one among $\{\eta_1(\mathbf{x}), \ldots, \eta_K(\mathbf{x})\}$, the classifier will yield a Bayes optimal prediction. For statistical learning model, such as the nearest neighbor classifier, $\Delta(x)$ can be viewed as a union bound that only grows with $O(\sqrt{\ln K})$. For deep neural networks, while it is hard to analyze the growth rate of $\Delta(\mathbf{x})$ over $K$ rigorously, such growth rate is slow by intuition and experience (Widmann et al., 2019; Zhao & Lai, 2022; Zhao et al., 2021; Kull et al., 2019; Rajaraman et al., 2022). As a result, the classification performance of our approach only decreases slowly with $K$.

Finally, we conduct experiments on both synthesized data and standard benchmark datasets. The goal of using synthesized data is to display how the performance degrades with increasing number of classes $K$. The results show that with small $K$, our method achieves nearly the same results with existing approaches. However, with $K$ increases, our vector approximation method exhibits clear advantages. We then test the performance on some standard benchmark datasets, which also validate the effectiveness of our proposed method.

The contributions of this work are summarized as follows.

- We propose a vector approximation approach for learning with label DP. Our method is convenient to implement and introduces little additional computational complexity.
- We provide a brief theoretical analysis showing that the model performance remains relatively stable with increasing number of classes $K$.
- Numerical results on synthesized data validate the theoretical analysis, which shows that the performance of our method only decays slightly with $K$. Experiments on real datasets also validate the effectiveness of our proposed method.

## 2 RELATED WORK

**Central label DP.** (Esfandiari et al., 2022) proposes an approach by clustering feature vectors, and then resampling each label based on the samples within the same cluster. Malek Esmaeili et al. (2021) applies PATE-FM (Papernot et al., 2017) into label DP settings, which splits the training datasets into several parts, and each part is used to train a teacher model. The labels are then altered by a majority vote from teacher models. Appropriate noise is added to the voting for privacy protection. Then PATE trains a student model using these altered labels. PATE is further improved in (Tang et al., 2022). Despite achieving good performances, these methods are designed under central label DP

requirement (see Definition 2), which requires that the curator is trusted, instead of protecting the privacy of each sample individually. Therefore, the accuracy scores are not comparable with ours.

**Local label DP.** Under local DP, labels are privatized before training to protect each sample individually. The performance of simple randomized response (Warner, 1965) drops with $K$ sharply. Here we list two main modifications:

*1) RRWithPrior (Ghazi et al., 2021).* This is an important improvement over the randomized response method, which partially alleviates the degradation of performance for large $K$. RRWithPrior picks transformed labels from some candidate labels with high conditional probabilities, instead of from all $K$ labels, thus it narrows down the list of candidates and somewhat reduces the label noise. However, this method requires multiple training stages to get a reliable estimate of prior probability. As a result, the samples in the beginning stages are used inefficiently. Moreover, RRWithPrior optimizes the probability of retaining the original label, i.e. $P(\tilde{Y} = Y)$, with $\tilde{Y}$ being the transformed one. Such optimization goal does not always align with the optimization of classification risk.

*2) ALIBI (Malek Esmaeili et al., 2021).* This method conducts one-hot encoding first and then adds Laplacian noise to protect privacy. The design shares some similarities with our method. For each sample, ALIBI generates a soft label vector using Bayesian inference and normalizes them to 1 using a softmax operation. However, as shown in Appendix B.2, ALIBI does not completely overcome the drawbacks of scalar approximation methods, since the soft label value still decreases with $K$. Moreover, the time complexity for transforming each label is $O(K^2)$, thus the computation cost is high with large $K$. On the contrary, our vector approximation only requires $O(K)$ time for each label transformation.

**Others.** There are also some works under other settings. (Ghazi et al., 2022; Badanidiyuru et al., 2023) study the regression problem under label LDP. Similar to the classification problem, these methods randomly perturb the label to another. Moreover, (Krichene et al., 2024; Chua et al., 2024; Ma et al., 2024; Ghazi et al., 2024) discussed the case with both public and private features. There are also some methods that deal with the drawback of label DP and propose to protect the privacy of training samples in a better way (Wu et al., 2023; Busa-Fekete et al., 2023). Furthermore, Wei et al. (2023) gives a comprehensive empirical evaluation of label DP algorithms, and Zhao et al. (2025) provides a theoretical analysis of the fundamental limits of learning with label differential privacy.

In this work, we observe that all methods based on scalar approximation have inherent limitation of information loss. In particular, the probability of retaining the original label after local randomization inevitably reduces with $K$ (see Appendix B.1). With the intuition of preserving more information, instead of transforming each label to another scalar label, our vector approximation approach privatizes each label into a vector $\mathbf{Z}$ that is not normalized to 1. Compared with existing methods, our approach introduces little additional computational complexity, while achieving better performance with large number of classes $K$.

## 3 PRELIMINARIES

### 3.1 LABEL DP

To begin with, we recall some concepts related to DP and introduce the notion of label DP.

**Definition 1.** *(Differential Privacy (DP) (Dwork et al., 2006)). Let $\epsilon \in \mathbb{R} \geq 0$. An algorithm $\mathcal{A}$ is $\epsilon$-differentially private (denoted as $\epsilon$-DP) if for any two adjacent datasets $D$ and $D'$ and any $S \subseteq \text{range}(\mathcal{A})$,*

$$P(\mathcal{A}(D) \in S) \leq e^{\epsilon} P(\mathcal{A}(D') \in S), \tag{1}$$

*in which two datasets $D$ and $D'$ are adjacent if they differ on a single training sample, including both the feature vector and the label.*

In supervised learning problems, the output of algorithm $\mathcal{A}$ is the trained model, while the input is the training dataset. Under Definition 1, both features and labels are privatized. However, in some scenarios, protecting features can be unnecessary, and we focus solely on the privacy of labels. Correspondingly, the notion of label DP is defined as follows:

**Definition 2.** *(Label Differential Privacy (Label DP) (Chaudhuri & Hsu, 2011)) An algorithm $\mathcal{A}$ is $\epsilon$-label differentially private (denoted as $\epsilon$-label DP) if for any two datasets $D$ and $D'$ that differ on the label of a single training sample, and any $S \subseteq \mathrm{range}(\mathcal{A})$, (1) holds.*

Compared with Definition 1, now the algorithm only needs to guarantee (1) when the label of a sample changes.

Throughout this paper, we consider the local DP setting, i.e. each label is transformed to a random variable (or vector) $\mathbf{Z} = M(Y)$ individually by a mechanism $M : \mathcal{Y} \to \mathcal{Z}$, in which $\mathcal{Y}$ denotes the set of all labels, while $\mathcal{Z}$ is the space of transformed variables or vectors. The label LDP is defined as follows.

**Definition 3.** *A random mechanism $M : \mathcal{Y} \to \mathcal{Z}$ is $\epsilon$-label LDP if for any $y, y' \in \mathcal{Y}$ and any $S \subseteq \mathcal{Z}$,*

$$P(M(Y) \in S | Y = y) \leq e^\epsilon P(M(Y) \in S | Y = y'). \tag{2}$$

Given $N$ training samples $(\mathbf{X}_i, Y_i)$, $i = 1, \ldots, N$, the mechanism $M$ transforms $Y_i$ to $\mathbf{Z}_i$, $i = 1, \ldots, N$. It can be easily shown that label LDP is a stronger privacy requirement compared with central label DP (Definition 2). To be more precise, as long as $M$ is local $\epsilon$-label DP, then any learning algorithms using $(\mathbf{X}_i, \mathbf{Z}_i)$, $i = 1, \ldots, N$ as training samples are central $\epsilon$-label DP.

## 3.2 CLASSIFICATION RISK

Now we describe the goal and the evaluation metric of the classification task. For a $K$-class classification problem, the goal is to build a classifier that predicts the label $Y \in \mathcal{Y} = [K]$, in which $[K] = \{1, \ldots, K\}$, given a feature vector $\mathbf{X} \in \mathcal{X} \subseteq \mathbb{R}^d$. In this work, we use $0 - 1$ loss for classification. For a classifier $\hat{Y} = c(\mathbf{X})$, in which $\hat{Y}$ is the predicted label, the risk is defined as the expectation of the loss function:

$$R[\hat{Y}] = \mathbb{E}[\mathbf{1}(\hat{Y} \neq Y)] = \mathrm{P}(c(\mathbf{X}) \neq Y). \tag{3}$$

Define $K$ functions, named $\eta_1, \ldots, \eta_K$, as the conditional class probability given a specific feature vector $\mathbf{x}$:

$$\eta_j(\mathbf{x}) = \mathrm{P}(Y = j | \mathbf{X} = \mathbf{x}), j = 1, \ldots, K. \tag{4}$$

If $\eta_k$ is known, then for any test sample with feature vector $\mathbf{x}$, the classifier can make the prediction be the class with maximum conditional probability. Therefore, the Bayes optimal classifier $c^*$ is

$$c^*(\mathbf{x}) = \arg\max_{j \in [K]} \eta_j(\mathbf{x}). \tag{5}$$

The risk corresponds to the ideal classifier $c^*$, called Bayes risk, is

$$R^* = \mathrm{P}(c^*(\mathbf{X}) \neq Y). \tag{6}$$

Bayes risk is the minimum risk among all possible classifiers. In reality, $\eta_k$ is unknown. Therefore, for a practical classifier that is trained using finite number of samples, there is some gap between its risk and the Bayes risk. Such gap $R - R^*$ is called the excess risk. Denote

$$\eta^*(\mathbf{x}) = \max_j \eta_j(\mathbf{x}) \tag{7}$$

as the maximum of $m$ regression functions. The next proposition gives an expression of the excess risk.

**Proposition 1.** *For any practical classifier $c : \mathcal{X} \to \mathcal{Y}$, the excess risk is*

$$R - R^* = \int (\eta^*(\mathbf{x}) - \mathbb{E}[\eta_{c(\mathbf{x})}(\mathbf{x})]) f(\mathbf{x}) d\mathbf{x}, \tag{8}$$

*in which $R^*$ is the Bayes risk, and $f$ is the probability density function (pdf) of feature vector $\mathbf{X}$, $\eta_{c(\mathbf{x})}(\mathbf{x})$ is just $\eta_j(\mathbf{x})$ with $j = c(\mathbf{x})$. Note that $c(\mathbf{x})$ is random due to the randomness of the training dataset. Therefore the expectation in (8) is taken over $N$ training samples.*

*Proof.* From (5) and (6), the Bayes risk is

$$R^* = \mathrm{P}(Y \neq c^*(\mathbf{X})) = \int \mathrm{P}(Y \neq c^*(\mathbf{x})|\mathbf{X} = \mathbf{x})f(\mathbf{x})d\mathbf{x} = \int (1 - \eta^*(\mathbf{x}))f(\mathbf{x})d\mathbf{x}. \tag{9}$$

The actual risk for a practical classifier is

$$R = \mathrm{P}(Y \neq c(\mathbf{X})) = \mathbb{E}\left[\int \left(1 - \eta_{c(\mathbf{x})}(\mathbf{x})\right) f(\mathbf{x})d\mathbf{x}\right]. \tag{10}$$

From (10) and (9),

$$R - R^* = \int (\eta^*(\mathbf{x}) - \mathbb{E}[\eta_{c(\mathbf{x})}(\mathbf{x})])f(\mathbf{x})d\mathbf{x}. \tag{11}$$

The proof is complete. □

## 3.3 NOTATIONS

Finally, we clarify the notations. Throughout this paper, $a \wedge b = \min(a, b)$, $a \vee b = \max(a, b)$. We claim that $a \lesssim b$ if there exists a constant $C$ such that $a \leq Cb$. Notation $\gtrsim$ is defined similarly. $\mathrm{Lap}(\lambda)$ denotes the Laplacian distribution with parameter $\lambda$, whose probability density function (pdf) is $f(u) = e^{-|u|/\lambda}/(2\lambda)$. Furthermore, we use capital letters to denote random variables, and lowercase letters to denote values.

## 4 THE PROPOSED METHOD

Here we present and explain our new method. Section 4.1 shows the basic idea. Section 4.2 gives an intuitive analysis. Finally, practical implementation with deep learning is discussed in Section 4.3.

### 4.1 METHOD DESCRIPTION

Instead of transforming a label to another scalar, we construct a random vector $\mathbf{Z} \in \{0, 1\}^K$, which can preserve more information than a random one-dimensional scalar. To be more precise, denote $Z(j)$ as the $j$-th element of $\mathbf{Z}$.

Given the original label $Y = j'$, let the conditional distribution of $Z(j)$ be

$$\mathrm{P}(Z(j) = 1|Y = j') = \begin{cases} \frac{e^{\frac{\epsilon}{2}}}{1+e^{\frac{\epsilon}{2}}} & \text{if} \quad j = j' \\ \frac{1}{1+e^{\frac{\epsilon}{2}}} & \text{if} \quad j \neq j', \end{cases} \tag{12}$$

and

$$\mathrm{P}(Z(j) = 0|Y = j') = \begin{cases} \frac{1}{1+e^{\frac{\epsilon}{2}}} & \text{if} \quad j = j' \\ \frac{e^{\frac{\epsilon}{2}}}{1+e^{\frac{\epsilon}{2}}} & \text{if} \quad j \neq j'. \end{cases} \tag{13}$$

It is ensured that conditional on $Y$, $Z(j)$ are independent for different $j$. The privacy mechanism described by (12) and (13) is a simple form of RAPPOR (Erlingsson et al., 2014). RAPPOR was analyzed in (Kairouz et al., 2016) as one of common frequency estimation methods under LDP. According to (Kairouz et al., 2016), RAPPOR has the optimal convergence rate. Our work shows that RAPPOR can be also used in the randomization step of label DP. The following theorem shows the DP property.

**Theorem 1.** *The privacy mechanism (12), (13) is $\epsilon$-label LDP.*

*Proof.* For any $j$, $j'$ and any $\mathbf{z} \in \{0, 1\}^K$, we have

$$\begin{aligned} \frac{\mathrm{P}(\mathbf{Z} = \mathbf{z}|Y = j)}{\mathrm{P}(\mathbf{Z} = \mathbf{z}|Y = j')} &= \Pi_l \frac{\mathrm{P}(Z(l) = z(l)|Y = j)}{\mathrm{P}(Z(l) = z(l)|Y = j')} \\ &= \frac{\mathrm{P}(Z(j) = z(j)|Y = j)\mathrm{P}(Z(j') = z(j')|Y = j)}{\mathrm{P}(Z(j) = z(j)|Y = j')\mathrm{P}(Z(j') = z(j')|Y = j')} \\ &\leq (e^{\frac{\epsilon}{2}})^2 = e^\epsilon, \end{aligned} \tag{14}$$

in which the first step holds since $Z(l)$ are conditional independent on $l$. The second step holds because changing $Y$ from $j$ to $j'$ does not affect the conditional distributions of elements of $\mathbf{Z}$ other than the $j$-th and the $j'$-th one, i.e. $\mathrm{P}(Z(l) = z(l)|Y = j) = \mathrm{P}(Z(l) = z(l)|Y = j')$ for $l \notin \{j, j'\}$. Therefore, we only need to bound the ratio of the probability mass of the $j$-th and $j'$-th element. The last step holds since (12) and (13) ensures that $\mathrm{P}(Z(j) = z(j)|Y = j)/\mathrm{P}(Z(j) = z(j)|Y = j') \leq e^{\epsilon/2}$. Therefore, the label LDP requirement in Definition 3 is satisfied. $\qquad\square$

Following above procedures, the labels of all $N$ training samples $Y_1, \ldots, Y_N$ are transformed into $N$ vectors $\mathbf{Z}_1, \ldots, \mathbf{Z}_N$. From (5), the optimal prediction is the class $j$ that maximizes $\eta_j(\mathbf{x})$. However, $\eta_j$ is unknown, and we would like to find a substitute. Towards this end, we train a model that can be either parametric (linear models, neural networks, etc.) or nonparametric ($k$ nearest neighbors, tree-based methods, etc.), to approximate $\mathbf{Z}_1, \ldots, \mathbf{Z}_N$. The model output can be expressed by

$$\mathbf{g}(\mathbf{x}) = (g_1(\mathbf{x}), \ldots, g_K(\mathbf{x})). \tag{15}$$

$g_j(\mathbf{x})$ denotes the $j$-th element of the output vector. With a good model and sufficient training samples, $g_j(\mathbf{x})$ approximates $\mathbb{E}[Z(j)|\mathbf{X} = \mathbf{x}]$. It is expected that picking $j$ that maximizes $\eta_j(\mathbf{x})$ is nearly equivalent to maximizing $g_j(\mathbf{x})$. Therefore, in the prediction stage, for any test sample whose feature vector is $\mathbf{x}$, the predicted label is

$$\hat{Y} = c(\mathbf{x}) := \arg\max_{j \in [K]} g_j(\mathbf{x}). \tag{16}$$

### 4.2 A Brief Analysis

Now we briefly explain why our new method works. Define

$$\tilde{\eta}_j(\mathbf{x}) := \mathrm{P}(Z(j) = 1|\mathbf{X} = \mathbf{x}). \tag{17}$$

It is expected that the model output $(g_1(\mathbf{x}), \ldots, g_K(\mathbf{x}))$ approximates $(\tilde{\eta}_1(\mathbf{x}), \ldots, \tilde{\eta}_K(\mathbf{x}))$. Ideally, $g_j(\mathbf{x}) = \tilde{\eta}_j(\mathbf{x})$, then from (16), $c(\mathbf{x}) = \arg\max_j \tilde{\eta}_j(\mathbf{x}) = \arg\max_j \eta_j(\mathbf{x}) = c^*(\mathbf{x})$ is the optimal prediction. However, in reality, there are some gaps between $(g_1(\mathbf{x}), \ldots, g_K(\mathbf{x}))$ and $(\tilde{\eta}_1(\mathbf{x}), \ldots, \tilde{\eta}_K(\mathbf{x}))$, thus the prediction may be suboptimal. To quantify the effect of approximation error on the excess risk, denote $\Delta(\mathbf{x})$ as the maximum approximation error for all classes, i.e.

$$\Delta(\mathbf{x}) = \max_{j \in [K]} |g_j(\mathbf{x}) - \tilde{\eta}_j(\mathbf{x})|. \tag{18}$$

Then we show the following theorem.

**Theorem 2.** *Denote $\eta_s(\mathbf{x})$ as the second largest conditional class probability given $\mathbf{x}$, i.e.*

$$\eta_s(\mathbf{x}) = \max_{j \neq c^*(\mathbf{x})} \eta_j(\mathbf{x}). \tag{19}$$

*Then*

$$R - R^* \leq \int \mathbb{E}\left[2\Delta_\epsilon(\mathbf{x})\mathbf{1}(\eta^*(\mathbf{x}) - \eta_s(\mathbf{x}) \leq 2\Delta_\epsilon(\mathbf{x}))\right] f(\mathbf{x})d\mathbf{x}, \tag{20}$$

*in which*

$$\Delta_\epsilon(\mathbf{x}) := \frac{e^{\frac{\epsilon}{2}} + 1}{e^{\frac{\epsilon}{2}} - 1}\Delta(\mathbf{x}). \tag{21}$$

*Proof.* From Proposition 1, the prediction is suboptimal if $c(\mathbf{x}) \neq c^*(\mathbf{x})$, which happens only if

$$g_{c(\mathbf{x})}(\mathbf{x}) \geq g_{c^*(\mathbf{x})}(\mathbf{x}). \tag{22}$$

From the definition of $\Delta(\mathbf{x})$ in (18), (22) indicates that

$$\tilde{\eta}_{c(\mathbf{x})}(\mathbf{x}) \geq \tilde{\eta}^*(\mathbf{x}) - 2\Delta(\mathbf{x}). \tag{23}$$

From (12), (13) and (17), for all $j \in [K]$,

$$
\begin{aligned}
\tilde{\eta}_j(\mathbf{x}) &= \sum_{l=1}^{K} \mathrm{P}(Y = l | \mathbf{X} = \mathbf{x}) \mathrm{P}(Z(j) = 1 | Y = l) \\
&= \frac{1}{1 + e^{\epsilon/2}} \sum_{l \neq j} \eta_l(\mathbf{x}) + \frac{e^{\epsilon/2}}{1 + e^{\epsilon/2}} \eta_j(\mathbf{x}) \\
&= \frac{1}{1 + e^{\epsilon/2}} + \frac{e^{\epsilon/2} - 1}{e^{\epsilon/2} + 1} \eta_j(\mathbf{x}),
\end{aligned}
\tag{24}
$$

in which the last step holds since from (4), $\sum_j \eta_j(\mathbf{x}) = 1$. Therefore, (23) is equivalent to

$$
\eta_{c(\mathbf{x})}(\mathbf{x}) \geq \eta^*(\mathbf{x}) - 2\Delta_\epsilon(\mathbf{x}),
\tag{25}
$$

in which $\Delta_\epsilon$ has been defined in (21). Note that for all $j \neq c^*(\mathbf{x})$, the value of $\eta_j(\mathbf{x})$ is at most $\eta_s(\mathbf{x})$ (recall the definition (19)). Therefore, a suboptimal prediction $c(\mathbf{x}) \neq c^*(\mathbf{x})$ can happen only if

$$
\eta^*(\mathbf{x}) - \eta_s(\mathbf{x}) \leq 2\Delta_\epsilon(\mathbf{x}).
\tag{26}
$$

(26) suggests that as long as the gap between the largest and the second largest conditional class probability is large enough with respect to $\epsilon$ and the function approximation error $\Delta(\mathbf{x})$, the classifier will make an optimal prediction, i.e. $\hat{Y} = c^*(\mathbf{x})$. Hence

$$
\eta^*(\mathbf{x}) - \eta_{c(\mathbf{x})}(\mathbf{x}) \leq \begin{cases} 2\Delta_\epsilon(\mathbf{x}) & \text{if} \quad \eta^*(\mathbf{x}) - \eta_s(\mathbf{x}) \leq 2\Delta_\epsilon(\mathbf{x}) \\ 0 & \text{if} \quad \eta^*(\mathbf{x}) - \eta_s(\mathbf{x}) > 2\Delta_\epsilon(\mathbf{x}). \end{cases}
\tag{27}
$$

Recall Proposition 1,

$$
\begin{aligned}
R - R^* &= \mathbb{E}\left[ \left( \eta^*(\mathbf{x}) - \mathbb{E}[\eta_{c(\mathbf{x})}(\mathbf{x})] \right) f(\mathbf{x}) d\mathbf{x} \right] \\
&\leq \mathbb{E}\left[ \int \mathbf{1}\left( \eta^*(\mathbf{x}) - \eta_s(\mathbf{x}) \leq 2\Delta_\epsilon(\mathbf{x}) \right) \cdot 2\Delta_\epsilon(\mathbf{x}) f(\mathbf{x}) d\mathbf{x} \right].
\end{aligned}
\tag{28}
$$

The proof of (20) is complete. $\qquad\square$

Here we interpret Theorem 2. As a union bound on the estimation error of all classes, $\Delta(\mathbf{x})$ usually only grows slightly with $K$. For example, in Appendix C, we show that for $k$ nearest neighbor classifiers, $\Delta(\mathbf{x})$ grows with $\sqrt{\ln K}$. Deep learning models are much harder to analyze rigorously (Guo et al., 2017), thus we only provide an intuitive argument in Appendix C to show that with increasing $K$, $\Delta$ remains nearly stable. Therefore, from (20), if the shape of $\eta^*(\mathbf{x}) - \eta_s(\mathbf{x})$ is nearly fixed, then the excess error will not grow significantly with $K$. This happens if $K$ is large but the conditional class probabilities given $\mathbf{x}$ for most classes are very small.

### 4.3 PRACTICAL IMPLEMENTATION

Finally, we discuss the practical implementation with deep learning. (15) contains $K$ models $g_1, \ldots, g_K$. Practically, it is not necessary to construct $K$ neural networks. For the convenience of implementation, we can let $\mathbf{w}_1, \ldots, \mathbf{w}_K$ share most parameters, except the last layer. Therefore, we only need to construct one neural network. The last layer can use sigmoid activation to match the privatized vector $\mathbf{Z}$ using binary cross entropy loss. This will simplify the implementation and avoid extra computation costs. Compared with non-private learning, our method only introduces $O(NK)$ additional time complexity for label transformation.

Here a sigmoid activation at the last layer can not be replaced by softmax, since the randomized vector $\mathbf{Z}$ does not sum to 1. Instead, $\mathbf{Z}$ may contain multiple elements with value 1, thus we need to use sigmoid activation so that the output $(g_1(\mathbf{x}), \ldots, g_K(\mathbf{x}))$ can approximate $(\tilde{\eta}_1(\mathbf{x}), \ldots, \tilde{\eta}_K(\mathbf{x}))$.

Now we summarize the differences between the randomized response method (including its modifications (Ghazi et al., 2021)) and our new approach. The differences are illustrated in Figure 1. Instead of randomly flipping the label and then conducting one-hot encoding, we use a random vectorization according to (12) and (13). The resulting vector may contain multiple elements whose values are 1. The activation of the last layer is also changed to sigmoid. Correspondingly, the loss function for model training is the binary cross entropy loss, instead of the multi-class version.

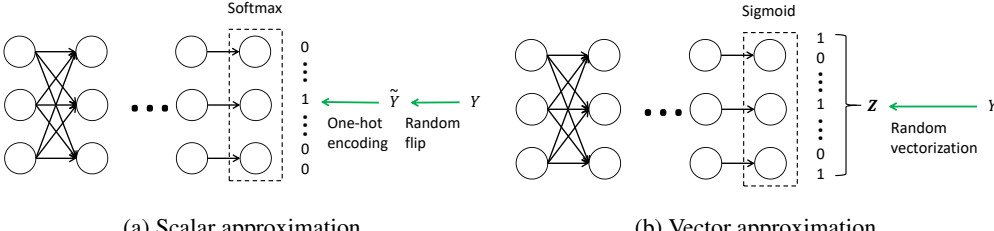

(a) Scalar approximation.         (b) Vector approximation.

Figure 1: An illustrative figure to compare randomized response and our new method.

# 5 NUMERICAL EXPERIMENTS

## 5.1 SYNTHESIZED DATA

To begin with, we show some results using randomly generated training samples to show how is the accuracy affected by increasing the number of classes $K$. Samples in all $K$ classes are normally distributed with mean $\mu_i$, $i = 1, \ldots, K$ and standard deviation $\sigma$. $\mu_i$ are set to be equally spaced on a unit circle. For all experiments, we generate $N = 10000$ training samples. We use the $k$ nearest neighbor method with $k = 200$ as the base learner of all models. We compare our method with the randomized response, LP-2ST (which uses RRWithPrior randomization) (Ghazi et al., 2021) and ALIBI (Malek Esmaeili et al., 2021). All methods satisfy the label LDP requirement with $\epsilon = 1$. The results are shown in Figure 2, in which each dot represents the average accuracy score over $1,000$ random trials.

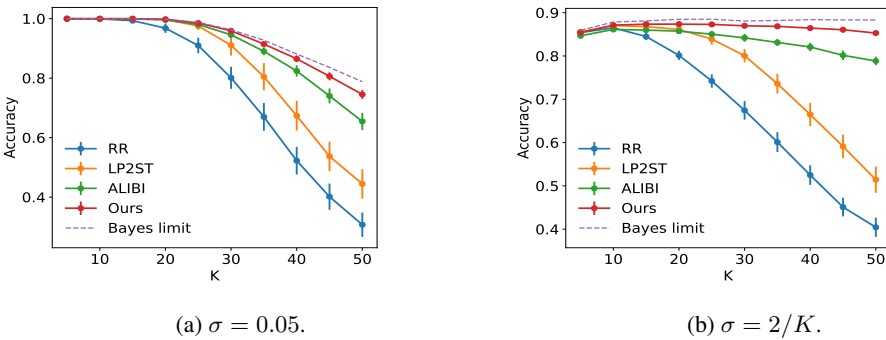

(a) $\sigma = 0.05$.            (b) $\sigma = 2/K$.

Figure 2: Comparison of the performances of methods of learning with label DP with varying number of classes $K$. The purple dashed line denotes the accuracy of Bayes optimal classifier (5).

Figure 2(a) shows the result using $\sigma = 0.05$. The result shows that with small $K$, our method achieves nearly the same accuracy as existing methods. However, with large $K$, the vector approximation method (the red line) shows clear advantage over other methods. Recall that our theoretical analysis shows that if $\eta^*(\mathbf{x}) - \eta_s(\mathbf{x})$ is fixed, the excess risk will not grow significantly with the number of classes $K$. However, from Figure 2(a), the accuracy of vector approximation still decreases with $K$. Our explanation is that with large $K$, there is a stronger overlap between distributions for neighboring classes, thus $\eta^*(\mathbf{x}) - \eta_s(\mathbf{x})$ becomes lower on average. According to Theorem 2, the excess risk is also larger correspondingly. In Figure 2(b), we show the result with $\sigma = 2/K$, so that $\eta^*(\mathbf{x}) - \eta_s(\mathbf{x})$ remains nearly the same with increasing $K$. In this case, with increasing $K$, the accuracy of our vector approximation approach remains stable.

In general, the experiment results with synthesized data validate our theoretical analysis and show that our method improves existing methods for cases with large number of classes $K$. Moreover, we show some additional experiments in Appendix A with other parameters, which validates that our method consistently outperforms existing approaches with large $K$.

## 5.2  REAL DATA

Now we evaluate our new method on standard benchmark datasets that have been widely used in previous works on differentially private machine learning, including MNIST (LeCun, 1998), Fashion MNIST (Xiao et al., 2017) CIFAR-10 and CIFAR-100(Krizhevsky et al., 2009). For the MNIST and Fashion MNIST datasets, we use a simple convolutional neural network composed of two convolution and pooling layers with dropout rate $0.5$, which achieves $99\%$ accuracy for non-private training samples. For CIFAR-10 and CIFAR-100, we use ResNet-18, which achieves $95\%$ and $76\%$ accuracy for non-private data, respectively. In recent years, many new models have emerged, such that the state-of-the-art accuracies on CIFAR-10 and CIFAR-100 have been greatly improved, such as vision transformer (Dosovitskiy et al., 2020). However, we still use ResNet-18 for a fair comparison with existing works. In our experiments, we set the batch size to be $400$, and use the Adam optimizer with learning rate $0.001$. Since our method is significantly different from previous ones in both label generation and loss functions, the optimal parameters of our method are different from those in existing methods. Therefore, we compare all methods with their own optimal parameters. Following conventions of existing works (Wei et al., 2022; De et al., 2022; Tramer & Boneh, 2021), we ignore the additional privacy cost caused by parameter tuning.

Table 1: Experiments on MNIST and Fashion MNIST dataset under different privacy levels using a simple CNN. The accuracy for non-private data is $99\%$.

| $\epsilon$ | 0.2 | 0.3 | 0.5 | 0.7 | 1.0 | 2.0 |
|---|---|---|---|---|---|---|
| RR | 58.0 | 64.1 | 74.9 | 89.5 | 95.3 | 97.7 |
| LP-2ST | 47.5 | 64.3 | 81.4 | 87.9 | 91.8 | 97.7 |
| ALIBI | 50.9 | 55.1 | 84.3 | 89.4 | 94.5 | 97.6 |
| **Ours** | **63.9** | **78.4** | **87.9** | **93.4** | **96.2** | **97.4** |

(a) MNIST

| $\epsilon$ | 0.5 | 1.0 | 1.5 | 2.0 |
|---|---|---|---|---|
| RR | 59.6 | 74.6 | 79.7 | 84.7 |
| LP-2ST | 60.1 | 75.6 | 82.4 | 85.0 |
| ALIBI | 54.3 | 79.8 | 80.1 | 84.9 |
| **Ours** | **75.7** | **83.4** | **84.7** | **85.9** |

(b) Fashion MNIST

Table 1 shows the results on MNIST and Fashion MNIST datasets. The number in the table refers to the accuracy, i.e. percentage of correct predictions. LP-2ST is the method in (Ghazi et al., 2021) using two stages. ALIBI comes from (Malek Esmaeili et al., 2021). The results show that our new method outperforms existing methods in general. With large $\epsilon$, such that the accuracy is already close to the non-private case, the accuracy may be slightly lower than existing methods due to some implementation details and some random factors.

Table 2: Experiments on CIFAR-10 dataset using ResNet-18. The accuracy for non-private data is $95\%$.

| $\epsilon$ | 1 | 2 | 3 | 4 |
|---|---|---|---|---|
| LP-1ST | 60.0 | 82.4 | 89.9 | 92.6 |
| LP-2ST | 63.7 | 86.1 | 92.2 | 93.4 |
| LP-1ST$\star$ | 67.6 | 84.0 | 90.2 | 92.8 |
| LP-2ST$\star$ | 70.2 | 87.2 | 92.1 | 93.5 |
| ALIBI | 71.0 | 84.2[1] | - | - |
| **Ours** | **75.1** | **91.6** | **92.7** | **93.2** |

Table 2 shows the result with the CIFAR-10 dataset. In the table, LP-1ST$\star$ and LP-2ST$\star$ refer to the LP-1ST and LP-2ST methods, respectively, in which models have been pretrained with the CIFAR-100 dataset, which is regarded as public and no privacy protection is needed. Such setting follows (Abadi et al., 2016). The results of the first four rows of Table 2 come from (Ghazi et al., 2021), Table 1. The fifth row is the result of ALIBI, which comes from (Malek Esmaeili et al., 2021), Table 1. For the CIFAR-10 dataset, our method outperforms existing approaches with small $\epsilon$. Note that our method does not use pre-training. However, the results are still better than existing methods even with pre-training. Similar to the experiment of the MNIST dataset, the advantage of our method

becomes less obvious when $\epsilon$ is large, and the accuracies of existing methods are already close to the non-private baseline.

Table 3 shows the result with the CIFAR-100 dataset. The results of LP-2ST and ALIBI come from Table 2 in (Ghazi et al., 2021) and (Malek Esmaeili et al., 2021), respectively. The results show that our method significantly outperforms LP-2ST and ALIBI. The advantage of our method is especially obvious with small $\epsilon$. It is worth mentioning that ALIBI calculates the soft label instead of conducting just a label flipping. The soft label can be expressed by a vector, thus the idea of ALIBI shares some similarity with our vector approximation method. As a result, ALIBI performs significantly better than LP-2ST. However, from Table 3, our method also outperforms ALIBI.

Table 3: Experiments on CIFAR-100 dataset using ResNet-18. The accuracy for non-private data is 76%.

| $\epsilon$ | 2 | 3 | 4 | 5 | 6 | 8 |
|---|---|---|---|---|---|---|
| LP-2ST | - | 28.7 | 50.2 | 63.5 | 70.6 | 74.1 |
| ALIBI | - | 55.0 | 65.0 | 69.0 | 71.5 | 74.4 |
| **Ours** | **64.4** | **67.7** | **71.8** | **73.8** | **74.1** | **74.4** |

Here we summarize the experimental results. The synthesized data shows that our vector approximation method has relatively stable performance with the increase of the number of classes $K$. As a result, with large $K$, our method has a significantly better performance compared with existing methods. Such observation agrees well with our theoretical analysis in Section 4.2. We have also tested the results using benchmark datasets, which validate our method in real scenarios. It worths mentioning that several works on central label DP achieve better accuracy than those reported here, such as PATE (Malek Esmaeili et al., 2021) and its improvement in (Tang et al., 2022). As a weaker privacy requirement compared with label LDP, these results under central label DP are not comparable to our results.

## 6 CONCLUSION

In this paper, we have proposed a new method for classification with label differential privacy based on vector approximation. The general idea is to generate a noisy random vector first to protect the privacy of labels and train the model to match the privatized vector. Compared with existing scalar approximation methods, which conduct label flipping and let the flipped label match the model outputs, our method has better theoretical convergence properties and empirical performances. In particular, our method solves the problem of the degradation of utility with large number of classes $K$. Our analysis suggests that the prediction is optimal as long as the gap between the largest and second largest conditional class probability is enough, thus the risk only grows slightly with $K$. Moreover, our method is easier to implement and only requires $O(NK)$ time for label privatization. Numerical experiments with both synthesized data and standard benchmark datasets validate the effectiveness of our proposed method.

## ACKNOWLEDGEMENT

The work of Li Shen is supported by STI 2030—Major Projects (No. 2021ZD0201405), Shenzhen Basic Research Project (Natural Science Foundation) Basic Research Key Project (NO. JCYJ20241202124430041), the Major Project in Judicial Research from Supreme People's Court (NO. GFZDKT2024C08-3), and CCF-DiDi GAIA Collaborative Research Funds (NO. CCF-DiDi GAIA 202419). The work of Zhikun Zhang is supported in part by the NSFC under Grants No. 62441618, 62402431, and Zhejiang University Education Foundation Qizhen Scholar Foundation. The work of Zhe Liu is supported by National Science Foundation China No.62132008.

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

## A  ADDITIONAL NUMERICAL EXPERIMENTS

Here we present some additional experiments.

**Higher** $\epsilon$. Figure 2 shows the result with $\epsilon = 1$. Now we set $\epsilon = 2$, and the results are shown in Figure 3.

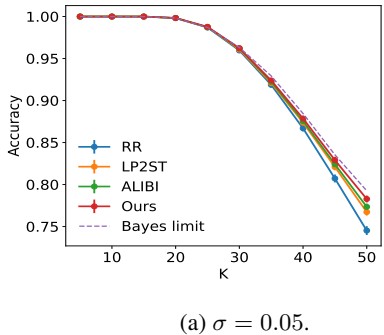
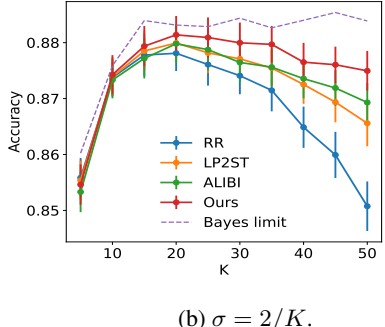

(a) $\sigma = 0.05$.

(b) $\sigma = 2/K$.

Figure 3: Experiments with simulated data with $\epsilon = 2$, $k = 100$.

With a weaker privacy requirement $\epsilon = 2$, the performances of all methods become better. The vector approximation method still outperforms existing methods.

**Other** $k$. Now we still use $\epsilon = 1$. However, recall that previous experiments use $k = 200$. Now we use $k$ nearest neighbor method with $k = 100$ as the base learner.

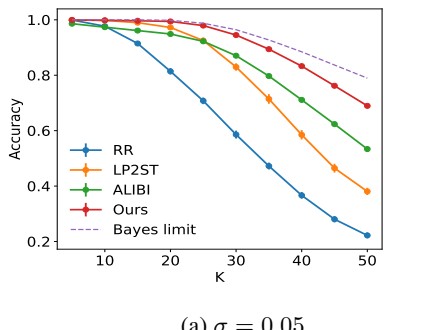 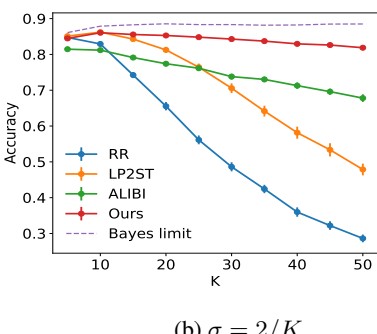

(a) $\sigma = 0.05$.  (b) $\sigma = 2/K$.

Figure 4: Experiments with simulated data with $\epsilon = 1$, $k = 100$.

## B  COMMENTS ON EXISTING METHODS

In this section, we show some negative results of existing methods, which indicate that the performance decreases with $K$ inevitably.

### B.1  SCALAR APPROXIMATION METHODS

Here we show that for all scalar approximation methods, which transform the label $Y$ to $Z$, the probability of retaining the original label decreases with $K$.

Denote $M_{ij} = P(Z = j|Y = i)$ as the transition probability. Then $M_{ii} = P(Z = i|Y = i)$ is the probability of retaining the original label $i$. The transition probability matrix is normalized, i.e. $\sum_{j=1}^{K} M_{ij} = 1$. From the $\epsilon$-label LDP constraint, $M_{ii} \leq e^{\epsilon} M_{ji}$ for all $j$. Taking sum over all $i$ and $j$, the following inequality holds:

$$K \sum_{i=1}^{K} M_{ii} \leq \sum_{i=1}^{K} \sum_{j=1}^{K} e^{\epsilon} M_{ji} = K e^{\epsilon}.$$

Therefore, the average probability of retaining original label satisfies $(1/K) \sum_{i=1}^{K} M_{ii} \leq e^{\epsilon}/K$, which decays inversely with $K$. The result indicates that for all scalar approximation methods, the performance degrades inevitably with larger $K$.

### B.2  ALIBI

ALIBI (Malek Esmaeili et al., 2021) generates a soft label for each input, which partially solves the problem caused by scalar approximation. The algorithm runs as follows. Firstly, let $\mathbf{o} = \text{OneHot}(Y) + W$, in which $W \sim \text{Lap}(\lambda)$. Then $\mathbf{o}$ is $\epsilon$-LDP with respect to $Y$, thus any learning algorithms based on any post-processing of $\mathbf{o}$ satisfies $\epsilon$-label LDP. If we assume uniform prior, then the soft label is generated as follows:

$$p(y = c|\mathbf{o}, \lambda) = \text{softmax}(f(o_c)/\lambda), \tag{29}$$

in which $f(o_c) = -\sum_k |\mathbf{o}_k - [c = k]|$, and $[\cdot]$ is the Iverson bracket.

Now we show that the performance of ALIBI decays with $K$ by a simple example. Suppose that for all $x$, $\eta_1(x) = 1$, $\eta_2(x) = \ldots = \eta_K(x) = 0$. Then all samples have label 1. According to (29), it can be shown that the probability of making a correct prediction is upper bounded by

$$p(Y = 1|\mathbf{o}, \lambda) \leq \frac{1}{\sum_{c=2}^{K} \mathbb{1}(f(o_c) \geq f(o_1))} \leq \frac{1}{\sum_{c=2}^{K} \mathbb{1}(o_c \geq o_1)}, \tag{30}$$

The second inequality holds because if $o_c \geq o_1$, then it can be shown that $f(o_c) \geq f(o_1)$.

Recall Algorithm 3 in ALIBI paper, $o_1 = 1 + W_1$, $o_c = W_c$ for $c = 2, \ldots, K$, in which $W_j \sim Laplace(\lambda)$ for all $j = 1, \ldots, K$. Therefore, in expectation, $\sum_{c=2}^{K} \mathbf{1}(o_c \geq o_1)$ grows linearly with $K$. As a result, $p(Y = 1 | o, \lambda)$ decays nearly inversely with $K$.

From the example above, it can be found that the performance of ALIBI decays significantly with $K$.

## C   FURTHER ARGUMENTS ABOUT APPROXIMATION ERROR

### C.1   $k$ NEAREST NEIGHBOR METHOD

Now we bound $\Delta(\mathbf{x})$ for $k$ nearest neighbor approach, which has been extensively analyzed in literatures (Chaudhuri & Dasgupta, 2014; Döring et al., 2018; Zhao & Lai, 2021)[2]. Recall that the output structure expressed in (15). For the kNN classifier, let

$$g_j(\mathbf{x}) = \frac{1}{k} \sum_{i \in \mathcal{N}_k(\mathbf{x})} Z_i(j), \tag{31}$$

in which $\mathcal{N}_k(\mathbf{x})$ is the set of $k$ nearest neighbors of $\mathbf{x}$ among $\{\mathbf{X}_1, \ldots, \mathbf{X}_N\}$.

Define $p_r(\mathbf{x}) = \int_{B(\mathbf{x},r)} f(\mathbf{u}) d\mathbf{u}$ as the probability mass of $B(\mathbf{x}, r)$, in which $f$ is the pdf of feature vector $\mathbf{X}$. Define

$$r_0(\mathbf{x}) = \inf \left\{ r \middle| p_r(\mathbf{x}) \geq \frac{2k}{N} \right\}. \tag{32}$$

Define $\rho(\mathbf{x}) = \max_{i \in \mathcal{N}_k(\mathbf{x})} \|\mathbf{X}_i - \mathbf{x}\|$ as the kNN distance. Assume that $\eta_j(\mathbf{x})$ is $L$-Lipschitz. Now we bound the approximation error as follows.

$$
\begin{aligned}
|g_j(\mathbf{x}) - \eta_j(\mathbf{x})| &= \left| \frac{1}{k} \sum_{i \in \mathcal{N}_k(\mathbf{x})} Z_i(j) - \tilde{\eta}_j(\mathbf{x}) \right| \\
&\leq \left| \frac{1}{k} \sum_{i \in \mathcal{N}_k(\mathbf{x})} (Z_i(j) - \tilde{\eta}_j(\mathbf{X}_i)) \right| + \left| \frac{1}{k} \sum_{i \in \mathcal{N}_k(\mathbf{x})} (\tilde{\eta}_j(\mathbf{X}_i) - \tilde{\eta}_j(\mathbf{x})) \right| \\
&\leq \left| \frac{1}{k} \sum_{i \in \mathcal{N}_k(\mathbf{x})} (Z_i(j) - \tilde{\eta}_j(\mathbf{X}_i)) \right| + \frac{e^{\epsilon/2} - 1}{e^{\epsilon/2} + 1} L \rho(\mathbf{x}),
\end{aligned}
\tag{33}
$$

in which the last step uses (24). From Hoeffding's inequality,

$$\mathrm{P}\left( \left| \frac{1}{k} \sum_{i \in \mathcal{N}_k(\mathbf{x})} (Z_i(j) - \tilde{\eta}_j(\mathbf{X}_i)) \right| > \sqrt{\frac{1}{2k} \ln \frac{2K}{\delta}} \right) \leq \frac{\delta}{K}. \tag{34}$$

Moreover, define $n(\mathbf{x}, r)$ as the number of samples in $B(\mathbf{x}, r)$, then

$$
\begin{aligned}
\mathrm{P}(\rho(\mathbf{x}) > r_0(\mathbf{x})) &\leq \mathrm{P}(n(\mathbf{x}, r_0(\mathbf{x})) < k) \\
&\leq e^{-N p_{r_0(\mathbf{x})}(\mathbf{x})} \left( \frac{eN p_{r_0(\mathbf{x})}(\mathbf{x})}{k} \right)^k \\
&= e^{-2k}(2e)^k = e^{-(1 - \ln 2)k}.
\end{aligned}
\tag{35}
$$

From (33), (34) and (35), consider the definition of $\Delta$ in (18), with probability at least $1 - \delta - e^{-(1 - \ln 2)k}$,

$$\Delta(\mathbf{x}) \leq \sqrt{\frac{1}{2k} \ln \frac{2K}{\delta}} + \frac{e^{\epsilon/2} - 1}{e^{\epsilon/2} + 1} L r_0, \tag{36}$$

---

[2]To avoid confusion, we emphasize that lowercase $k$ is the number of nearest neighbor used in the kNN method, while $K$ is the number of classes.

which grows with $\sqrt{\ln K}$.

Finally, we discuss the convergence of $\Delta(\mathbf{x})$ (and the excess risk) with respect to $N$. With fixed $k$, according to (36), $\Delta(\mathbf{x})$ does not converge to zero even if $N \to \infty$. Therefore, we need to let $k$ grows with $N$ with an appropriate speed. We now focus on the case with $\epsilon < 1$. In (36) in our paper, assuming that the density is bounded away from zero, then from (32), $r_0 \sim \epsilon(k/N)^{1/d}$. Therefore, to minimize the right hand side of (36), we can let $k \sim \epsilon^{-2d/(d+2)} N^{2/(d+2)}$ (here we neglect logarithmic factors), then $\Delta(x)$ decays with $\epsilon^{d/(d+2)} N^{-1/(d+2)}$. According to (20) and (21), we have $\Delta_\epsilon(x) \sim (N\epsilon^2)^{-1/(d+2)}$. Therefore $R - R^* \lesssim (N\epsilon^2)^{-1/(d+2)}$.

In addition, in nonparametric classification, a commonly used assumption is "Tsybakov margin condition (Tsybakov, 2009; Audibert & Tsybakov, 2007)", which says that $P(0 < \eta^*(X) - \eta_s(X) < t) \lesssim t^\gamma$ for some $\gamma$. In this case, the convergence rate of the excess risk becomes faster, $R - R^* \lesssim (N\epsilon^2)^{-(1+\gamma)/(d+2)}$.

## C.2 NEURAL NETWORK

Neural networks are much harder to analyze compared with traditional methods like kNN. Here we just provide an argument to intuitively show that the approximation $g_j(\mathbf{x}) \approx \tilde{\eta}_j(\mathbf{x})$ holds.

In deep neural network models, the parameters can be expressed as $\mathbf{w} = (\mathbf{w}_1, \ldots, \mathbf{w}_K) \in \mathcal{W}_1 \times \ldots \times \mathcal{W}_K$, and the output (15) is now $g_j(\mathbf{x}) = h_j(\mathbf{w}_j^*, \mathbf{x})$, in which $h_j$ is the output of $j$-th neural network model. $\mathbf{w}_j^*$ is trained using $(\mathbf{X}_i, \mathbf{Z}_i)$, $i = 1, \ldots, N$ to minimize the empirical loss:

$$\mathbf{w}_j^* = \arg\min_{\mathbf{w}_j \in \mathcal{W}_j} \frac{1}{N} \sum_{i=1}^N l(h_j(\mathbf{w}_j, \mathbf{X}_i), Z_i(j)), j \in [K], \tag{37}$$

in which $l$ is the surrogate loss function used for training. A typical choice of $l$ is the binary cross entropy loss.

Let $\mathcal{G}_j = \{h_j(\mathbf{w}_j, \cdot) | \mathbf{w}_j \in \mathcal{W}_j\}$ be the set of all possible functions $g_j$ parameterized by $\mathbf{w}_j$. Then from (37),

$$\frac{1}{N} \sum_{i=1}^N l(g_j(\mathbf{w}_j^*, \mathbf{X}_i), Z_i(j)) = \min_{u \in \mathcal{G}_j} \frac{1}{N} \sum_{i=1}^N l(u(\mathbf{X}_i), Z_i(j)). \tag{38}$$

Denote $\mathcal{G} = \{g : \mathcal{X} \to [0, 1]\}$ as the set of all functions mapping from $\mathcal{X}$ to $[0, 1]$. Then

$$
\begin{aligned}
\mathbb{E}[l(g_j(\mathbf{w}_j^*, \mathbf{X}), Z(j))] \quad &\stackrel{(a)}{\approx} \quad \min_{u \in \mathcal{G}_j} \mathbb{E}[l(u(\mathbf{X}), Z(j))] \\
&\stackrel{(b)}{\approx} \quad \min_{u \in \mathcal{G}} \mathbb{E}[l(u(\mathbf{X}), Z(j))] \\
&= \quad \mathbb{E}\left[\min_{u \in \mathcal{G}} \mathbb{E}[l(u(\mathbf{x}), Z(j)) | \mathbf{X} = \mathbf{x}]\right].
\end{aligned}
\tag{39}
$$

In (39), (a) holds because from weak law of large number, the right hand side of (38) converges to its expectation. For the left side of (38), theories about generalization has been widely established (Mohri et al., 2018). The generalization gap converges to zero with the increase of sample size $N$. (b) uses the universal approximation theorem (Barron, 1993; Liang & Srikant, 2016; Yarotsky, 2018). For a neural network that is sufficiently wide and deep, each function in $\mathcal{G}$ can be closely approximated by a function in $\mathcal{G}_j$.

From now on, suppose that $l$ is the binary cross entropy loss, i.e. $l(q, p) = -p \ln q - (1 - p) \ln(1 - q)$. It can be shown that $\tilde{\eta}_j(\mathbf{x})$ is the minimizer of $\mathbb{E}[l(u(\mathbf{x}), Z(j)) | \mathbf{X} = \mathbf{x}]$. Recall that $\mathcal{G}$ is defined as the set of all functions mapping from $\mathcal{X}$ to $[0, 1]$. Hence, $g_j(\mathbf{w}_j^*) \in \mathcal{G}$. This implies

$$\mathbb{E}\left[l(g_j(\mathbf{w}_j^*, \mathbf{X}), Z(j))\right] \geq \mathbb{E}\left[\min_{u \in \mathcal{G}} \mathbb{E}[l(u(\mathbf{x}), Z(j)) | \mathbf{X} = \mathbf{x}]\right]. \tag{40}$$

From (39), the left hand side of (40) is not larger than the right hand side too much. Therefore, there exists a $\delta > 0$ such that

$$\mathbb{E}\left[\min_{u \in \mathcal{G}} \mathbb{E}[l(u(\mathbf{x}), Z(j)) | \mathbf{X} = \mathbf{x}]\right] \leq \mathbb{E}\left[l(g_j(\mathbf{w}_j^*, \mathbf{X}), Z(j))\right] \leq \mathbb{E}\left[\min_{u \in \mathcal{G}} \mathbb{E}[l(u(\mathbf{x}), Z(j)) | \mathbf{X} = \mathbf{x}]\right] + \delta. \tag{41}$$

Recall the steps in (39). $\delta$ goes to zero with the increase of model complexity and sample size $N$.

Assume that $g_j(\mathbf{w}_j^*, \mathbf{x})$ is continuous in $\mathbf{x}$. Then $l(g_j(\mathbf{w}_j^*, \mathbf{x}), Z(j))$ is also continuous in $\mathbf{x}$. As a result, $l(g_j(\mathbf{w}_j^*, \mathbf{x}), Z(j))$ can not be much larger than $\min_{u \in \mathcal{G}} \mathbb{E}[l(u(\mathbf{x}), Z(j))|\mathbf{X} = \mathbf{x}]$, otherwise there will be a neighborhood of $\mathbf{x}$ called $\mathcal{N}(\mathbf{x})$, such that $l(g_j(\mathbf{w}_j^*, \mathbf{x}'), Z(j))$ is much larger than $\min_{u \in \mathcal{G}} \mathbb{E}[l(u(\mathbf{x}'), Z(j))|\mathbf{X} = \mathbf{x}']$ for all $\mathbf{x}' \in \mathcal{N}(\mathbf{x})$, which violates the second inequality in (41). Therefore,

$$\mathbb{E}[l(g_j(\mathbf{w}_j^*, \mathbf{x}), Z(j))|\mathbf{X} = \mathbf{x}] \approx \min_{u \in \mathcal{G}} \mathbb{E}[l(u(\mathbf{x}), Z(j))|\mathbf{X} = \mathbf{x}]. \tag{42}$$

Recall that $l$ denotes the binary cross entropy loss, thus

$$l(u(\mathbf{x}), Z(j)) = -Z(j) \ln u(\mathbf{x}) - (1 - Z(j)) \ln(1 - u(\mathbf{x})). \tag{43}$$

Recall the definition of $\tilde{\eta}_j$ in (17), we have

$$\mathbb{E}[l(u(\mathbf{x}), Z(j))] = -\tilde{\eta}_j(\mathbf{x}) \ln u(\mathbf{x}) - (1 - \tilde{\eta}_j(\mathbf{x})) \ln(1 - u(\mathbf{x})). \tag{44}$$

To minimize $\mathbb{E}[l(u(\mathbf{x}), Z(j))|\mathbf{X} = \mathbf{x}]$, we have $u(\mathbf{x}) = \tilde{\eta}_j(\mathbf{x})$. Therefore, from (42),

$$g_j(\mathbf{x}) = h_j(\mathbf{w}_j^*, \mathbf{x}) \approx \tilde{\eta}_j(\mathbf{x}). \tag{45}$$

Such argument holds for all $j = 1, \ldots, K$. Recall (18), $\Delta(\mathbf{x})$ can be obtained by a union bound of $g_j(\mathbf{x}) - \tilde{\eta}_j(\mathbf{x})$.

