# OpenReview forum: "Enhancing Learning with Label Differential Privacy by Vector Approximation"
_ICLR.cc/2025/Conference — ICLR 2025 Spotlight_

### Official Review · Reviewer_nJ8J · 2024-10-30

**Soundness:** 3
**Presentation:** 2
**Contribution:** 3
**Rating:** 8
**Confidence:** 4

**Summary:**

This paper introduces a vector approximation method aimed at improving label differential privacy (label DP) for large multi-class datasets. By encoding labels as vectors rather than scalars, the authors argue that this approach retains more class information, enabling better performance in high-class settings where traditional label DP methods degrade significantly. They present both theoretical analysis and empirical validation on synthesized and benchmark datasets, demonstrating superior accuracy, especially as the number of classes $K$ grows.

**Strengths:**

- The paper addresses a meaningful problem: label LDP learning with a large number of label classes. This has important practical relevance, particularly in applications with long-tailed data distributions.
- The paper proposes a simple yet effective technique to tackle this issue.
- The theoretical results are adaptable and could be extended to various scenarios under specific assumptions (e.g., smooth regression functions).
- The experimental results demonstrate significant improvements over previous methods.

**Weaknesses:**

- The theoretical results provide only an upper bound for the proposed method. I suggest including additional comparisons and discussions between this bound and existing results. Additionally, could the authors establish any lower bounds for previous methods that use scalar labels alone (and possibly find the phase transition between $\log K$ and $K$ dependence)? Such bounds would clearly demonstrate the necessity of the proposed method.
- The experiments were conducted on standard datasets. It would be valuable for the authors to conduct experiments on datasets with long-tailed distributions to highlight the method's practical relevance. Furthermore, it would be interesting to discuss how a long-tail distribution or class imbalance might impact the theoretical guarantees of the algorithm.
- The paper is somewhat repetitive and could benefit from further editing for conciseness.

**Questions:**

- What is the purpose of the footnote in Table 3?

Also, see weaknesses.

---

> ### Author Response · Authors · 2024-11-14
> **Response**
>
> Thanks for your feedback. We are encouraged that you have a positive evaluation on the significance, as well as the simplicity and effectivenss of our work.
>
> **The theoretical results provide only an upper bound for the proposed method. I suggest including additional comparisons and discussions between this bound and existing results.**
>
> We think that this is a very good question. Deriving the lower bound of excess risk is hard. However, it can be shown that if we just use scalar labels, the probability of retaining original label decrease polynomially with $K$.
>
> Denote $M_{ij}=P(Z=j|Y=i)$ as the transition probability. Then $\sum_{j=1}^K M_{ij} = 1$. From the $\epsilon$-label LDP constraint, $M_{ii}\leq e^\epsilon M_{ji}$ for all $j$. Taking sum over all $i$ and $j$,
>
> $$K\sum_{i=1}^K M_{ii} \leq \sum_{i=1}^K \sum_{j=1}^K e^\epsilon M_{ji} = Ke^\epsilon.$$
>
> The average probability of retaining original label satisfies $(1/K)\sum_{i=1}^K M_{ii}\leq e^\epsilon/K$, which decays inversely with $K$.
>
> Therefore, with the increase of $K$, the privatized scalar random variables become increasingly unlikely to retain the original label.
>
> We thank you for the other two valuable suggestions. In our revised paper, we will add experiments with long-tailed distributions or datasets with high class imbalance. The expected result is that classes with minor probability will not obviously affect the performance of our method, but baseline methods will be negatively affected. We will also improve the conciseness.
>
> For the question, Table 3 is about experimentson on CIFAR-100 dataset. It seems that this table does not contain footnotes, thus we are not sure where you are referring to. Would you please explain the question further?

---

> > ### Comment · Reviewer_nJ8J · 2024-11-14
> >
> > I think the rebuttal addresses an interesting point. Hence, after reading the response and other reviewer's comments, I decide to raise my score.

---

### Official Review · Reviewer_DCbg · 2024-10-31

**Soundness:** 3
**Presentation:** 3
**Contribution:** 3
**Rating:** 6
**Confidence:** 3

**Summary:**

This paper introduces a new (?) approach for local label differential privacy (LDP) in machine learning, presenting both the theoretical framework and a numerical evaluation of the proposed method’s effectiveness. This method is based on a component-wise randomised response mechanism on the one-hot encoding of the class.

**Strengths:**

- The paper is well-organized and easy to follow, helping readers grasp the main ideas and methodology.
- The authors provide a numerical evaluation of the proposed methods, which helps to demonstrate its performance in practice.
- The authors provide a semi formal justification of the performance of the proposed method

**Weaknesses:**

- The focus on local differential privacy is not clear from the title and abstract, which could be misleading for readers. Revising these sections to clearly indicate this focus would improve accessibility and transparency.
- The methods mentioned in the “Others” section of the related work could be discussed in more detail to give readers a clearer understanding of the techniques used in prior studies.
- The practical applicability of the method is limited, as it requires training a separate discriminant for each class, which could be cumbersome or impractical for larger or more complex tasks.
- The novelty of the proposed method is questionable, as noted by the authors themselves in lines 232-233, where they acknowledge its prior use in the literature.
- The authors claim that their method achieves better results by encoding more information in a vector rather than a scalar, but it remains unclear if this added information genuinely benefits the privacy-utility tradeoff. It would be valuable to see a theoretical result similar to Theorem 2 applied to the randomized response mechanism over the simplex to support a comparative analysis.

**Questions:**

Could the authors provide justifications or address the weaknesses mentioned above, especially regarding the novelty of the method and the theoretical basis of the claimed improvements? If these concerns are adequately addressed, I would be open to revising my rating and overall assessment of the paper.

---

> ### Author Response · Authors · 2024-11-16
> **Response**
>
> Thanks for your review.
>
> **1.The focus on local differential privacy is not clear from the title and abstract, which could be misleading for readers. Revising these sections to clearly indicate this focus would improve accessibility and transparency.**
>
> Thanks for this comment. We will change the title to "Enhancing learning with local label differential privacy...". We will also mention it in the abstract.
>
> **2.The methods mentioned in the “Others” section of the related work could be discussed in more detail to give readers a clearer understanding of the techniques used in prior studies.**
>
> Actually, the related works mentioned in "others" section are not comparable baselines. They are under different settings. Ghazi et al. (2022) and Badanidiyuruetal.(2023) study the regression problem instead of classification. Krichene et al.(2024)  and Chua et al.(2024)
> are about learning problems with some features being public. In our revised paper, we will briefly mention these methods and discuss the relationship between these works and ours.
>
> **3. The practical applicability of the method is limited, as it requires training a separate discriminant for each class, which could be cumbersome or impractical for larger or more complex tasks.**
>
> This is a misunderstanding. The practical applicability is not limited. We do not require training a separate discriminant for each class. We mention it only for theoretical analysis, which gives an idea why our method outperforms existing methods, especially for large $K$.
>
> On the contrary, as mentioned by reviewer zGeU and nJ8J, our method is simpler than existing methods, instead of being cumbersome or impractical. We do not train the model multiple times as in Ghazi et al. 2021. Moreover, the label randomization also takes less time (only O(K) for each sample).
>
> **4. The novelty of the proposed method is questionable, as noted by the authors themselves in lines 232-233, where they acknowledge its prior use in the literature.**
>
> As mentioned in lines 232-233, RAPPOR was already proposed before. However, this method was used for frequency estimation, which is different from learning with label DP. Before our work, it was not obvious that RAPPOR can be used into learning with label DP.
>
> In our work, we apply an existing technique for one task to another different task, and achieve significant improvement over existing methods for this new task. We believe that this should be considered novel. The simplicity of our method and the connection with existing methods for other tasks should be an additional merit, instead of a lack of novelty.
>
> **5. The authors claim that their method achieves better results by encoding more information in a vector rather than a scalar, but it remains unclear if this added information genuinely benefits the privacy-utility tradeoff. It would be valuable to see a theoretical result similar to Theorem 2 applied to the randomized response mechanism over the simplex to support a comparative analysis.**
>
> Thanks for this valuable comment. We refer to the response to reviewer nJ8J.
>
> Deriving the lower bound of the excess risk is hard. However, it can be shown that the probability of retaining the original label decrease significantly with $K$, as long as we use scalar to encode information. Our analysis indicates that the average probability can not exceed $e^\epsilon/K$. Therefore, this added information indeed enhances the privacy-utility tradeoff.

---

> > ### Comment · Reviewer_DCbg · 2024-11-22
> >
> > Thank you for your detailed response and for clarifying the misunderstanding regarding the practical applicability of your method. I appreciate the thoughtful revisions you plan to make to the title, abstract, and related work sections. Seeing the enthusiasm of other reviewers and with a better understanding of your contributions, I have decided to raise my rating from 3 to 6.

---

### Official Review · Reviewer_Vdjs · 2024-11-03

**Soundness:** 4
**Presentation:** 4
**Contribution:** 3
**Rating:** 8
**Confidence:** 4

**Summary:**

This paper introduces a new approach to local label differential privacy, transforming each label into a K-dimensional vector, where each component is independently randomized with a RAPPOR-like mechanism. This vector-based approach preserves more information than scalar label transformations while maintaining ε-local differential privacy. The authors provide theoretical analysis showing that their method's performance only degrades slightly with increasing number of classes K. They empirically validate their approach using both synthetic data and standard benchmark datasets (MNIST, Fashion-MNIST, CIFAR-10, CIFAR-100), demonstrating that their method significantly outperforms existing approaches, especially when the number of classes is large and the privacy requirements are strict (small ε). Notably, proposed method outperforms a similar approach proposed by Malek Esmaeili et al. (2021), also relying on expanding a label to a vector of probabilities and adding noise to the vector.

**Strengths:**

1. The paper studies an important and well defined problem of label DP, which however does not get as much attention in the literature as traditional DP, while being relevant for many practical applications
2. The paper proposes a novel method to provide local label-only DP guarantees, and thoroughly explores it. The proposed method provides better utility (at the same privacy budget) compared to the prior methods on standard benchmarks. The paper provides necessary proofs of privacy guarantees, as well as general intuition behind the method.
3. The paper does a good job of putting this work in the context of the overall field, providing important details of prior works (notably, by Malek Esmaeili et al. (2021) as it shares the most similarities with the current paper)
4. The evaluation protocol looks solid and follows standard practices and baselines, including what is typically used by prior works.
5. The proposed practical implementation is quite neat, making the implementation of the method computationally efficient.

**Weaknesses:**

At it's core, the proposed method can be summarised as applying technique from RAPPOR (Erlingsson et al. (2014)) to the "soft label" approach to label DP introduced by ALIBI (Malek Esmaeili et al. (2021)). Both techniques are properly acknowledged by the authors, and combining two existing techniques in a novel way is not a weakness by itself. Especially given the strong empirical results and accompanying theoretical analysis.

However, I believe this should affect the focus of the analysis. In particular, the main focus area for the theoretical part of the paper is answering the question "how does the excess risk grow with the number of classes K", implicitly comparing it with the randomized response and RRWithPrior (Ghazi et al. (2021)), which are known to degrade in utility with growing K. Given strong conceptual similarities between ALIBI and the method proposed here, as well as the fact that the former is the current state-of-the-art for label DP task, I would have expected the theoretical analysis to focus on comparing the two methods. Or, if the analysis of $\Delta(x)$ is relevant for comparison with ALIBI, it needs to be shown.

Additionally, citing computational efficiency ($O(K^2$) vs $O(K$)) when comparing the new approach with ALIBI (Malek Esmaeili et al. (2021)) looks misplaced, when in any practical ML application the cost of training would dominate over the computational cost of flipping labels.

**Questions:**

1. Can you elaborate your reasoning on the line 107: "However, ALIBI generates outputs using Bayesian inference and normalizes them to 1 using a softmax operation. Intuitively, such normalization causes bias"?
2. In section 4.3 Practical Implementation, how does changing the activation function of the last layer affects the model performance? Do you use this in the experiments chapter and if yes, how do you modify the training process?
3. To be pedantic, line 87 should say "Malek Esmaeili et al. (2021) proposes PATE-FM". Original PATE was introduced by [Papernot et al., 2016](https://arxiv.org/abs/1610.05755), and is not specific to label DP.

---

> ### Author Response · Authors · 2024-11-14
> **Response**
>
> Thanks for your positive evaluation and valuable comments.
>
> # Reply to weaknesses
>
> **1. Given strong conceptual similarities between ALIBI and the method proposed here, as well as the fact that the former is the current state-of-the-art for label DP task, I would have expected the theoretical analysis to focus on comparing the two methods.**
>
> We think that this is a really good question. To make a rigorous theoretical statement, we need to show the **lower bound** of the risk of ALIBI. ALIBI uses Bayesian inference, which is a bit complex and it is thus hard to obtain a general lower bound. However, we can show that the performance of ALIBI decays with $K$ by a simple example.
>
> Suppose that for all $x$, $\eta_1(x)=1$, $\eta_2(x)=\ldots=\eta_K(x)=0$. Then all samples have label 1. According to eq.(3) in the ALIBI paper,
>
> $$p(Y=c|o, \lambda) = Softmax(f(o_c)/\lambda),$$
>
> in which we assume uniform prior. Then it can be shown that the probability of making a correct prediction is upper bounded by
>
> $$p(Y=1|o, \lambda)\leq \frac{1}{\sum_{c=2}^K 1(f(o_c)\geq f(o_1))}\leq \frac{1}{\sum_{c=2}^K 1(o_c\geq o_1)},$$
>
> in which $f(o_c)=-\sum_k |o_k-[c=k]|$, with $[]$ being the Iverson bracket. These are from the original paper of ALIBI. The second inequality holds because if $o_c\geq o_1$, then it can be shown that $f(o_c)\geq f(o_1)$.
>
> Recall Algorithm 3 in ALIBI paper, $o_1=1+W_1$, $o_c=W_c$ for $c=2,\ldots, K$, in which $W_j\sim Laplace(\lambda)$ for all $j=1,\ldots, K$. Therefore, in expectation, $\sum_{c=2}^K \mathbf{1}(o_c\geq o_1)$ grows linearly with $K$. As a result, $p(Y=1|o, \lambda)$ decays nearly inversely with $K$.
>
> From the example above, it can be found that the performance of ALIBI decays significantly with $K$.
>
> Hope that the above example provides some intuition. However, currently it is hard to provide a complete analysis since deriving the lower bound of risk is usually much harder than the upper bound.
>
> **2.citing computational efficiency ($O(K^2)$ vs $O(K)$) when comparing the new approach with ALIBI (Malek Esmaeili et al. (2021)) looks misplaced, when in any practical ML application the cost of training would dominate over the computational cost of flipping labels.**
>
> In this paper, we discuss general methods for label randomization under local label DP. The base learner is not necessarily a deep learning model. It can also be statistical machine learning methods. Therefore we think that it is worthwhile to mention the complexity. For example, in our numerical experiments using $k$ nearest neighbors, the computation cost of ALIBI is about 3-4 times higher than our method.
>
> # Reply to questions
>
> **1. Can you elaborate your reasoning on the line 107**
>
> Thanks for this comment. The softmax operation normalizes the output to 1, which makes the values of each component smaller if the previous sum was larger than 1, or larger if the previous sum is smaller than 1, thus this operation introduces bias. For simplicity, suppose $o_1=1$ and $o_2=\ldots=o_K=0$, then $f(o_1)=0$ and $f(o_2)=-2$. According to eq.(3) in ALIBI paper, after softmax operation, the probability $p(y=1|o,\lambda)<1$, while $p(y=c|o, \lambda)>0$ for $c\neq 1$. This is an example that the softmax operation introduces bias.
>
> However, we agree that readers may get confused when reading this sentence at the first time. Therefore in the revised paper, we will restate this sentence, and mention the example in the reply of weakness 1.
>
> **2. In section 4.3 Practical Implementation, how does changing the activation function of the last layer affects the model performance? Do you use this in the experiments chapter and if yes, how do you modify the training process?**
>
> The changing of activation function is required by our analysis. We hope that $g_j(x)$ approximates $E[Z(j)|X=x]$ (see the discussion after eq.(15) in our paper). Therefore, here we need to use the sigmoid activation. If we use softmax activation, then the network learns $E[Z(j)|X=x]/\sum_l E[Z(l)|X=x]$.
>
> **3. To be pedantic, line 87 should say "Malek Esmaeili et al. (2021) proposes PATE-FM". Original PATE was introduced by Papernot et al., 2016, and is not specific to label DP.**
>
> Thanks for this comment. We will change the statement and cite the original paper of Papernot et al.

---

> > ### Comment · Reviewer_Vdjs · 2024-11-25
> >
> > I thanks the authors for responding to my questions and concerns.
> > I light of these comments, I've decided to raise my score further to 8.

---

### Official Review · Reviewer_zGeU · 2024-11-04

**Soundness:** 4
**Presentation:** 3
**Contribution:** 3
**Rating:** 8
**Confidence:** 4

**Summary:**

This paper consider the problem of supervised multi-class learning problem under the setting of "label differential privacy" (LDP). This is a setting that is well-studied in recent literature, where the input features are considered public, and only the label is considered sensitive and needs to be protected by differential privacy.

This paper proposes a new randomizing mechanism that satisfies LDP, and outperforms prior methods studied in literature. The new method is as follows: Suppose there are $K$ possible labels. Given any label $y \in [K]$, construct the $K$-dimensional one-hot encoding of $y$ and apply the (binary) Randomized Response mechanism on each bit of the one-hot encoding, flipping each bit with probability $\frac{1}{1 + e^{\epsilon/2}}$.

A theoretical analysis is done for the k-nearest neighbor learning rule showing that the excess risk grows as $\sim \sqrt{\ln K}$ with increasing $K$ (keeping other parameter fixed).

Experimental results show that this approach outperforms prior methods studied in literature on a synthetic dataset (mixture of Gaussians data with centers around a circle) as well as real data (MNIST, CIFAR).

**Strengths:**

The paper proposes a simple method and shows that it outperforms prior methods proposed in literature of LP-MST (Label Privacy Multi Stage Training [[Ghazi et al. '21](https://arxiv.org/abs/2102.06062)]) and ALIBI [[Malek et al. '21](https://arxiv.org/abs/2106.03408)].

The method is also simple to implement with minimal computational overhead relative to prior methods.

The paper is well written and easy to follow.

**Weaknesses:**

I feel there is rooom for making the paper stronger on the theoretical side.

I feel the analysis of k-NN is a bit incomplete (Appendix B.1) and doesn't highlight the complete picture. What I would have liked to see is how the excess risk decreases as $N \to \infty$ (see more details about this under "Questions").

Nevertheless, I find it quite interesting that such a simple method shows a clear improvement over prior methods in the experiments, and so I don't think of the above as a significant weakness, since the theoretical analysis is not critical for the experimental results.

**Questions:**

In particular, [Ghazi et al. '21](https://arxiv.org/abs/2102.06062) have an analysis in Appendix F where they show that the excess risk of learning with stochastic gradient descent is $K / \sqrt{N}$ for pure-DP and $\sqrt{K / N}$ for approximate-DP. A subsequent paper by [Ghazi et al. '24](https://arxiv.org/abs/2406.19040) provides a more complex method that can theoretically achieve a excess empirical error rate of $\sqrt{(\log K) / N}$.

But I think the $\sim \sqrt{(\log K) / k}$ scaling does not exactly say how the excess error scales with $N$. I would imagine that as $N \to \infty$, the optimal learning rule would also be scaling $k$ in some way, and not keep it fixed. I think this analysis might bring in dimensionality factors into the picture, which are necessary for understanding the precise scaling of excess risk with $N$.

---

On the experimental side, I am wondering if the authors have also considered datasets with non-uniform distribution over labels. For example, one could consider MNIST or CIFAR datasets by additional duplicates of certain labels to make the distribution of labels to be biased. How does the proposed method perform against prior methods, especially methods that use prior information (such as RRWithPrior or LP-MST)?

---

> ### Author Response · Authors · 2024-11-16
> **Response**
>
> Thank you very much for your positive review. We are encouraged that you have a positive evaluation on the simplicity and effectiveness of our proposed method, as well as the clarity of our paper.
>
> In what follows, we reply to the weaknesses and questions you have raised.
>
> # Reply to Weaknesses
>
> **1.I feel there is rooom for making the paper stronger on the theoretical side.**
>
> Thanks for this comment. According to other reviewer's comment (Vdjs and nJ8J), it would be better to include the analysis of all methods based on scalar approximation. Moreover, we will also provide an analysis of ALIBI.
>
> **2.I feel the analysis of k-NN is a bit incomplete (Appendix B.1) and doesn't highlight the complete picture. What I would have liked to see is how the excess risk decreases as $N\rightarrow \infty$.**
>
> This is indeed an important point. With fixed $k$, the excess risk does not decrease to zero even if $N\rightarrow \infty$. Therefore, we need to let $k$ grows with $N$ with an appropriate speed. We now focus on the case with $\epsilon < 1$. In eq.(36) in our paper, assuming that the density is bounded away from zero, then $r_0\sim \epsilon(k/N)^{1/d}$ (see also eq.(32)). Therefore, to minimize the right hand side of eq.(34), we can let $k\sim \epsilon^{-2d/(d+2)}N^{2/(d+2)}$ (here we neglect logarithmic factors), then $\Delta(x)$ decays with $\epsilon^{d/(d+2)} N^{-1/(d+2)}$. According to eq.(20) and eq.(21), we have $\Delta_\epsilon(x)\sim (N\epsilon^2)^{-1/(d+2)}$. Therefore $R-R^*\lesssim (N\epsilon^2)^{-1/(d+2)}$.
>
> In addition, in nonparametric classification, a commonly used assumption is "Tsybakov margin condition", which says that $P(0<\eta^*(X)-\eta_s(X)<t)\lesssim t^\gamma$ for some $\gamma$. In this case, the convergence rate of the excess risk becomes faster, $R-R^*\lesssim (N\epsilon^2)^{-(1+\gamma)/(d+2)}$.
>
> In the revised paper, we have added the discussions in Appendix B.
>
> # Reply to Questions
>
> **About Ghazi et al's two papers.**
>
> As explained in the response to weakness 2, we will add the convergence with respect to $N$. However, the result is not comparable with Ghazi et al. 2021 and Ghazi et al. 2024. These two papers consider the stochastic optimization problem, thus the risk is the optimization risk instead of the classification risk. The optimization excess risk does not include the approximation error, i.e. the gap between the risk of current model (under optimal parameters) with the minimum risk over all models. If the hypothesis space of a learning model does not contain the ground truth, then the approximation error is nonzero.
>
> **On the experimental side, I am wondering if the authors have also considered datasets with non-uniform distribution over labels. For example, one could consider MNIST or CIFAR datasets by additional duplicates of certain labels to make the distribution of labels to be biased. How does the proposed method perform against prior methods, especially methods that use prior information (such as RRWithPrior or LP-MST)?**
>
> Thanks for this comment. We will run more experiments with non-uniform distribution of labels.

---

> > ### Comment · Reviewer_zGeU · 2024-11-22
> > **Acknowledgement**
> >
> > I thank the authors for the response. I will keep my rating.

---

### Author Response · Authors · 2024-11-21
**Global Response**

We thank reviewers for taking the time to read the paper as well as your valuable comments. We are encouraged that reviewers agree that our method is simple and effective (zGeU, Vdjs and nJ8J). We also notice that there are some feedbacks that help us to further improve our paper. According to these comments, we have revised our paper. All changes are marked in **blue** color.

The revision of this paper includes the following aspects:

1. According to Reviewer nJ8J's comment, we have included the negative result of all scalar methods (in Appendix B), which indicates that the probability of retaining the original label inevitably decreases with $K$. Therefore vector approximation method is necessary.

2. According to Reviewer Vdjs's comment, we have added the negative result of ALIBI, which shows that the soft label value of the ground truth label decreases with $K$. Therefore ALIBI does not completely overcome the drawback of scalar approximation methods.

3. We have updated several places where we make some discussions with existing works.

4. We have corrected some minor typos and issues.

Moreover, we notice that both reviewer zGeU and nJ8J mentioned that it would be helpful to conduct experiments with class imbalanced data. We agree that this will further validate our proposed approach. We are currently running these experiments and will add these results when these experiments are complete.

Hope that this global response, as well as individual responses below, can clear up any confusion. We are looking forward to further discussions and reevaluation of our paper.

---

### Meta-Review · Area_Chair_QHy2 · 2024-12-14

**Metareview:**

## Summary of Contributions

This paper studies label differential privacy (label-DP) setting, where we want to train an ML model and the examples are public but the labels are sensitive and should be protected with differential privacy. The paper proposes a simple but effective algorithm for randomizing the labels and training: Encode the labels as one-hot encoding and flip each bit with certain probability, and train normally with cross-entropy loss. Note that the label randomization is the same as the so-called RAPPOR algorithm (Erlingsson et al., 2014) but this is the first time it is used in the context of label-DP. Despite the relatively simple algorithm, the authors show that this consistently performs better than previous algorithms (Ghazi et al., NeurIPS'21; Malek et al., NeurIPS'21) which are more complicated and computational intensive. Furthermore, the authors provide some theoretical justifications for nearest-neighbor classifiers.

## Strengths

- The algorithm is very simple and has little overhead compared to non-private training. As a result, it can be very practical.

- Empirically, the algorithm consistently outperforms the aforementioned previously known algorithms on all datasets as long as the $\epsilon$ value is not too large. The improvements are especially large (sometimes 10+ percent) in the small $\epsilon$ (aka high-privacy) regime, which is arguably the most important regime for privacy protection.

- The authors also provide theoretical justifications.

## Weaknesses

- I don't see any major weakness here. A minor weakness is that the theoretical justification is given in the form of nearest-neighbor classifier, which is a different setting than previous work (e.g. Ghazi et al.'s theoretical results are for convex ERM). This means that it is not possible to directly compare the new theoretical results with existing ones.

## Recommendation

Given the algorithm is simple and clearly better (both from computational efficiency and model accuracy perspectives) than previous algorithms in label-DP, I think this paper is a clear accept to ICLR.

**Additional Comments On Reviewer Discussion:**

During the rebuttal, the authors clarify some confusing regarding their theoretical results for nearest-neighbor classifiers. They also add a comparison to the aforementioned Malek et al.'s work, which gives a clear example of the differences between the two methods.

---

### Decision · Program_Chairs · 2025-01-22

Accept (Spotlight)